

# Stress profile influences learning approach in a marine fish

Vincent Raoult, Larissa Trompf, Jane E. Williamson and Culum Brown

Department of Biological Sciences, Macquarie University, Australia

## ABSTRACT

The spatial learning skills of high and low stress juvenile mulloway (*Argyrosomus japonicus*) were tested in a dichotomous choice apparatus. Groups of fish were formed based on background blood cortisol levels and required to learn the location of a food reward hidden in one of two compartments. Low stress fish characterised by low background levels of the stress hormone cortisol had higher activity levels and entered both rewarded and unrewarded rooms frequently. Within the first week of exposure, however, their preference for the rewarded room increased, indicative of learning. Fish that had high background levels of cortisol, in contrast, showed low levels of activity but when they chose between the two rooms they chose the rewarded room most often but showed less improvement over time. After 12 days in the apparatus, both low and high stress fish had similar ratios of rewarded vs unrewarded room entrances. Our results suggest that proactive coping styles may increase exposure to novel contexts and thus favour faster learning but at the cost of reduced initial accuracy.

## INTRODUCTION

Learning is of key importance to mobile animals existing in heterogeneous environments and influences just about every aspect of their biology (*Brown, Laland & Krause, 2011*). Factors that affect learning may influence both an animal's ability to learn and/or the way in which it learns, for example via trade-offs between speed and accuracy (*Wang et al., 2015*) or through preferences for socially mediated versus private information (*Webster et al., 2013*). Personality, defined as consistent differences in behaviour between individuals across time and/or context (*Wolf & Weissing, 2012*), is one factor that may drive variability in learning approach. Personality traits such as boldness or shyness have been shown to be correlated to learning ability for a range of taxa (see *Griffin, Guillette & Healy, 2015*) including fish (*Trompf & Brown, 2014*; *Sneddon, 2003*).

Learning plays a key role in fish behaviour in both wild and captive populations (*Kieffer & Colgan, 1992*; *Brown, Laland & Krause, 2011*). In an aquaculture context, for example, captive-reared fish may have to learn to anticipate various husbandry related events such as cleaning, or how to interact with self-feeders. Groups of rainbow trout (*Oncorhynchus mykiss*) took approximately 25 days to learn to use self-feeders and there was considerable individual variation in efficacy (*Alanärä, 1996*). Historically much of this variation was thought to be explained by hierarchy, but variation in personality plays an underlying

Corresponding author
Vincent Raoult,
vincent.raoult@newcastle.edu.au

role in determining hierarchy (*Colléter & Brown, 2011)*. Thus, much of this variation in captive fish behaviour may also be due to differences in personality. Moreover, research is increasingly directed towards behavioural conditioning of hatchery-reared fish as a mean of diminishing mortality rates post release (*Kellison, Eggleston & Burke, 2000*; *Fairchild & Howell, 2004*; *Donadelli et al., 2015*; *Sloychuk, Chivers & Ferrari, 2016*). Much of this conditioning involves learned responses to predators and prey (*Brown & Laland, 2011*; *Brown & Day, 2002*). Thus, understanding the link between personality and learning ability has direct applications for fisheries, aquaculture and hatchery management.

The relationship between personality and learning may be mediated in part through an animal's stress physiology (*Raoult et al., 2012b*; *Thomson et al., 2011*). Individuals within a species may consistently vary in their behavioural and physiological response to stressful situations both between and within populations—correlations that have been characterised as coping styles (*Koolhaas et al., 1999*; *Øverli, Sørensen & Nilsson, 2006*). Proactive strategies are typified by high levels of activity, including active avoidance of stressful stimuli, and are associated with activation of the sympathetic-adrenomedullary system. Reactive coping styles, in contrast, are generally associated with high levels of passivity and immobility, and activation of the pituitary-adrenocortical system (the hypothalamus-pituitary-interrenal (HPI) axis in teleost fish) (*Øverli et al., 2007*; *Schjolden et al., 2006*). Proactive individuals tend to be more reliant on routine than reactive individuals. Reactive individuals, in contrast, show greater behavioural flexibility in response to changes in their environment (*Coppens, De Boer & Koolhaas, 2010*; *Mesquita, Borcato & Huntingford, 2015*). Stress and divergent coping styles have been linked to memory formation in fish. Rainbow trout selected for low stress responsiveness retained a learned conditioned response for a greater length of time compared to trout selected for high responsiveness (*Moreira, Pulman & Pottinger, 2004*). Moreover, *Barreto, Volpato & Pottinger (2006)* found that treating rainbow trout with cortisol-releasing implants impaired their memory processes. While a low stress response may be advantageous in an aquaculture setting (*Huntingford & Adams, 2005*; *Huntingford et al., 2010*), associated behavioural characteristics such as boldness may enhance learning.

Mulloway *Argyrosomus japonicus* (Temminck & Schlegel, 1844) are a schooling fish found along the eastern and southern shores of Australia. This species can grow to over 40 kg and is an aggressive predatory fish that is popular with game fishermen. Due to the species' rapid rate of growth, it is being considered for potential as an aquaculture species for both re-stocking and commercial uses in Australia (*Fielder, Allan & Bardsley, 1999*; *Guy, Mcilgorm & Waterman, 2014*). Links between boldness and blood cortisol concentrations in this species have been shown previously (*Raoult et al., 2012b*) but the relationship between personality and learning ability has yet to be investigated. Here we compared the learning ability of shy, high stress and bold, low stress juvenile mulloway in a simple spatial learning task where a food reward was hidden in one of two compartments. We predicted that bolder, low stress individuals should be faster to learn novel tasks than shy, high stress individuals. We also predicted that bolder fish may initially show higher levels of inaccuracy relative to shy fish.

## MATERIALS AND METHODS

Juvenile mulloway ($n = 150$) were obtained from aquaculture pens in Botany Bay, Australia, and transported to the experimental facilities in Cronulla's Fisheries Research Centre (CFRC) in Sydney. Subjects ($369 \pm 13$ mm length; $495 \pm 48.5$ g weight) were derived from the same spawning event with multiple parents, originating from brood stock collected from Botany Bay. These fish were the same individuals used in *Raoult et al. (2012b)*.

Fish were housed in an outdoor 5,000 L flow-through tank taking water from the adjacent bay, thus water temperature varied seasonally (mean of 14 °C) and the tank was exposed to natural variations in diurnal light and temperature. Individuals were fed ~2% of their weight in 0.5 mm commercial pellet feed daily.

### Pit tags and blood sampling

One month following transfer to experimental facilities, mulloway were anaesthetised in a light solution of AQUI-S (5 ml/1,000 L), gently netted from their housing tank using a soft mesh net, placed in a secondary surgical anaesthesia bath in an AQUI-S solution (10 ml/1,000 L) until they lost their buoyancy control, held with a wet towel, and fitted with Passive Integrated Transponder tags (PIT) as outlined in (*Raoult, Brown & Williamson, 2012a*). Trovan ID100 tags ($2.2 \times 11$ mm) were implanted via sterile needle inserters in the body cavity posterior to the pectoral fins. Simultaneously, blood was collected from the caudal vein (1–2 ml) for cortisol analysis. Resulting incisions were sealed using superglue to aid in tag retention and reduce haemorrhaging (*Raoult, Brown & Williamson, 2012a*). Each fish was placed in a highly aerated 100 L tank for post-handling recovery. Fresh seawater was then flushed through their gills using a low-pressure hose and the fish was monitored until recovery. The entire process from capture to recovery took less than 5 min per fish. The netting process caused minimal disturbance to the rest of the fish in the holding tank, although there was a slight increase in baseline cortisol levels over the tagging period which we controlled for statistically (see *Raoult et al., 2012b*). Tagging incisions fully healed in the first week.

Stress levels were determined by obtaining background blood cortisol levels from blood samples taken shortly after pit tagging. Note the tagging procedure was too rapid for the fish to mount a blood cortisol response and thus the levels detected reflect pre-existing variation in the population, for example through individual position in the hierarchy. Background blood cortisol levels were of interest as they likely represent natural conditions similar to those fish experienced in the housing tanks and during learning trials. We could not determine cortisol concentrations from water samples (*Scott & Ellis, 2007*; *Zuberi, Ali & Brown, 2011*) due to limitations in the aquaria facilities and the number of fish housed in the system. Blood samples were analysed for cortisol concentrations with a coat-a-count kit from Diagnostic Products Corporation (Los Angeles, CA, USA) (refer to *Raoult et al., 2012b* for further details). There was a positive relationship between sampling order and measured cortisol levels, however, this relationship only explained 6% of the variation, and a subsequent ANCOVA taking into account sampling order found no relationship between sampling order and stress type, suggesting that our cortisol assays were accurate assessments of fish background stress levels (*Raoult et al., 2012b*). Fish blood cortisol concentrations

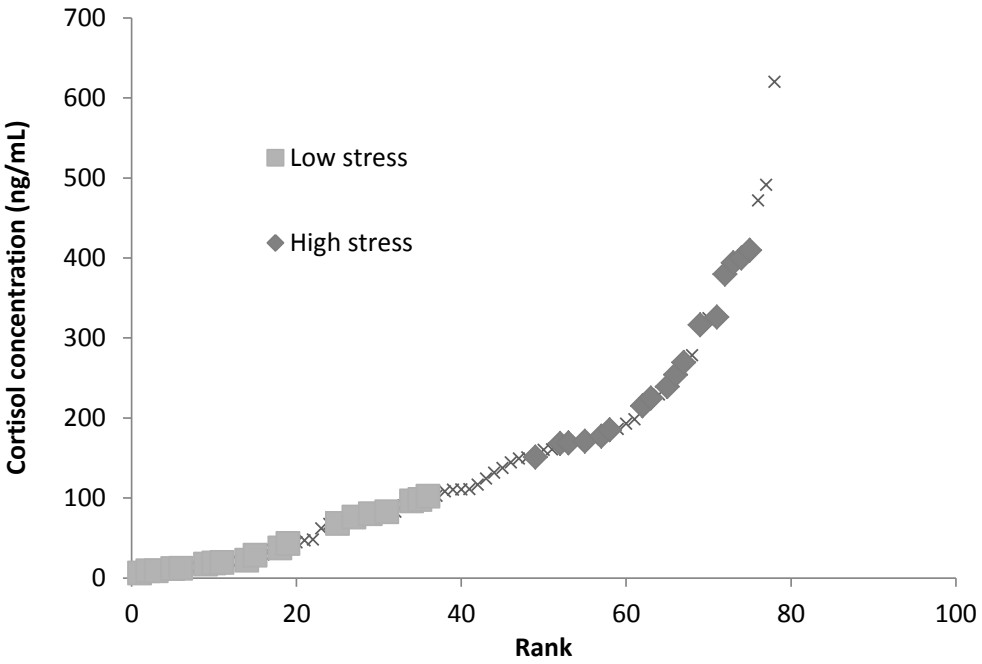

**Figure 1** **Ranked blood cortisol concentrations measured for entire sampled population of mulloway in this study.** Low-stress and high-stress fish used in this study are indicated with squares or diamonds, the remainder of the population is marked with *Xs*. Fish were separated into low-stress and high-stress depending on whether they were below or above the median blood cortisol concentration (110 ng/ml).

above the median blood cortisol concentration (110 ng/mL) were labelled as "high stress fish" (mean cortisol concentration of $261.7 \pm 22.25$ ng/ml) and those with concentrations below the median were labelled as "low stress fish" (mean cortisol concentration of $43.9 \pm 9.04$ ng/ml; Fig. 1). Eighteen low stress and eighteen high stress subjects were selected from either end of this distribution to take part in the learning trial (Fig. 1).

As growth rates are known to vary with coping style (*Biro et al., 2006*; *Øverli et al., 2007*), different growth rates over the lives of these aquaculture-grown mulloway could have resulted in size differences between low and high stress individuals, which would also lead to lower foraging activity (*Polverino et al., 2016*). However, the fish were all the same age and there was no significant correlation between body weight and cortisol concentration ($F_{1,76} = 2.34$, $p > 0.05$), and due to the relatively short duration of the study (<3 months) it was unlikely that growth rate of individuals influenced these experiments. Note, we made no attempt to quantify growth rate during the brief study period.

No mortalities were recorded during the study, from their retrieval in the aquaculture pens to the conclusion of research. Fish were retained in the aquarium facilities at CFRC following the end of the project. This research was conducted under Macquarie University Animal Ethics and Care approval number 2009/008.

### Learning test

Learning trials commenced four months after tag implantation. A spatial learning task required the fish to locate a food reward in one of two compartments. The experimental
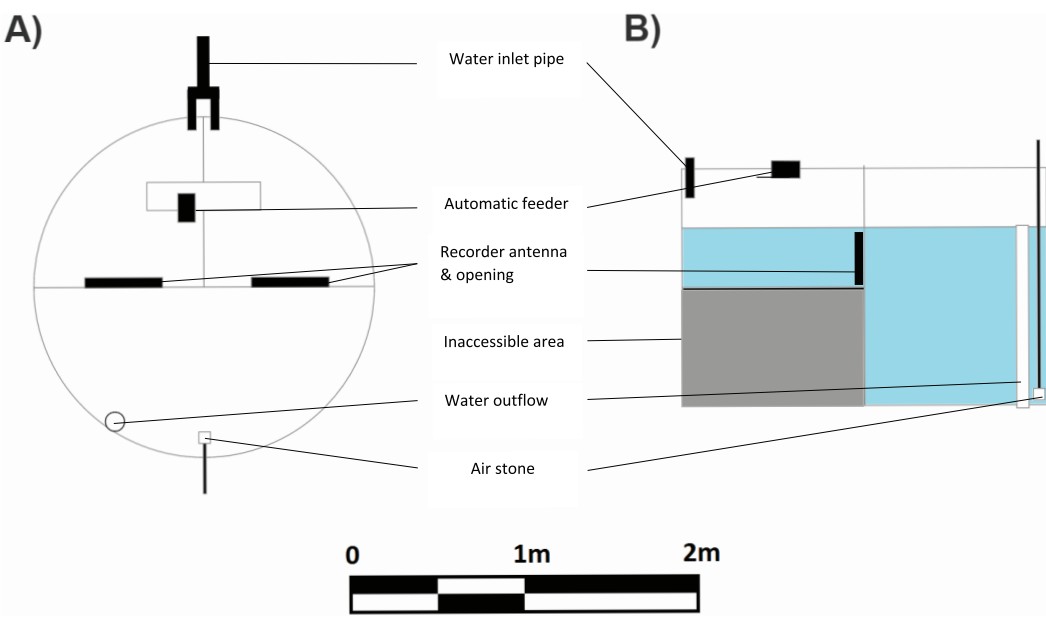

**Figure 2** Schematic (A) top view and (B) side view of experimental tank used for learning experiments. The learning experiment tank was a 2.4 m diameter and 1.2 m high flow-through system. The presence of PIT tag recorders at the entries of the rewarded and unrewarded compartments allowed a measure of activity into and out of those compartments for each individual fish.

arena used for the learning task consisted of a large circular tub (2.4 ø, 1.2 m deep) divided into three compartments using 2 mm thick opaque plastic sheets. Two smaller compartments were of equal size (ca 1/4 of the arena) and connected to the third large compartment (ca 1/2 the arena) via 27 cm circular doorways cut into the plastic dividers just below the water level (Figs. 2A, 2B). A false floor was created using breathable plastic weed mats in the two smaller compartments to reduce the depth to 40 cm. This encouraged the fish to enter the compartment and return to the larger, deeper compartment once they had finished eating. Pilot studies showed that these fish had a clear preference for the deeper compartment. The entrances to the small compartments were fitted with an ANT C270 antenna. LID650 PIT tag readers were attached to the antennae to record the time that individuals passed from one compartment to another. Preliminary tests conducted with the PIT tags alone verified that the readers would only record a tag when it was within the circular antenna.

An auto feeder (Eheim) was mounted above one of the small compartments and deposited feed at a set time each day as described below. Remaining food particles were siphoned off every morning. An inlet pipe was placed in each of the small compartments to maintain a high flow of water (10 L min$^{-1}$ each) through the system. The large compartment that housed the overflow and air stones was placed directly opposite the other two compartments.

Fish were tested in groups because *A. japonicus* does not respond well to social isolation (*Pirozzi, Booth & Pankhurst, 2009*). Six groups of six low or high stress individuals were tested (three groups of each stress type; $n = 36$ fish in total). While measuring the behaviour

of groups with similar stress characteristics could have led to an exaggeration of low-stress and high-stress response behaviours, this was preferable to mixing stress types, which could result in negative interactions between low-stress and high-stress individuals within groups that could dampen both behavioural types. Using mixed stress type groups would be more similar to natural conditions (*Verbeek, Iwamoto & Murakami, 2008*), while groups with a homogenous stress type would resemble aquaculture breeding conditions (*Castanheira et al., 2015*). Individuals were gently netted from their home tank, identified by their PIT tag and placed in the experimental tank for a 24 h acclimation period. Data recording began thereafter. As mulloway are reportedly largely nocturnal, the shoals were fed via the automatic feeders four times nightly (1900, 2200, 0100, 0400 h) with rations of 0.25% of their collective body. Trials were run for three weeks or until 2,000 PIT tag recordings were made for each experimental shoal, whichever occurred sooner. The 2,000 recording limit was chosen to ensure that the on-board memory of the recording apparatus would be sufficient to record all data.

## Analysis

The number of detections at the entrance to each of the small compartments was used to quantify the number of visits each fish made to each compartment each day. The data were analysed in three main ways. Firstly, we examined the change in global activity levels by examining the number of daily visits to both compartments for every fish over the length of the experiment. We also examined the change in the number of visits to the rewarded and unrewarded compartments separately. Our expectation was that low stress fish would be more active than high stress fish, but high stress fish might gradually habituate to the test area and thereby increase their activity levels. Secondly, we determined if low and high stress fish showed significant differences in the number of detections between the rewarded and unrewarded compartments over time. That is, did the fish develop a preference for the rewarded location and make fewer errors by avoiding the unrewarded location. The expectation was that fish would initially choose a compartment at random but would gradually learn which compartment contained the food reward. We hypothesised that low stress fish might show faster learning because their high activity levels lead them to rapidly explore the test arena. The total number of entrances into the rewarded compartment for each fish each day was log-transformed prior to analysis in all cases. The final analysis examined the ratio of detections in the rewarded vs unrewarded compartments for each individual each day. This analysis controlled for differences in total activity of the two groups of fish. If the fish learnt the location of the food reward, we expected the ratio of rewarded vs unrewarded compartment entries to increase over time. The resulting number between 0 and 1 was then adjusted by 0.5 so that a value of 0.5 indicated no preference for either compartment.

Data were analysed using general linear mixed models using groups as a random variable and stress status (high or low cortisol values) as a fixed variable. We employed a repeated measures approach with day number as the repeated measure. Data were analysed using R version 3.3.2 and the lme4 package. A restricted likelihood ratio test (RLRT) was used to determine whether the random factor group had an effect with the RLRsim

**Table 1 Summary of generalized linear model of recordings per day into the rewarded and unrewarded compartments for low stress and high stress responding fish.**

| Factor | Estimate | Standard error | t value | P value |
|---|---|---|---|---|
| Day | 0.14 | 0.02 | 6.44 | <0.001 |
| Stress type | 0.32 | 0.23 | 1.42 | 0.16 |
| Compartment | −1.68 | 0.11 | −15.88 | <0.001 |
| Stress type * Day | 0.09 | 0.03 | 2.94 | <0.01 |

package. Preference results were transformed using a logit transformation, and numbers of recordings were log + 1 transformed. Interaction effects between day and stress type were measured for recording models to record differences in activity over time, but not for preference models since preference levels should start and end at similar optimums for both treatments (start with no preference for either compartment and end with similar 'learnt' preference). In addition, adding an interaction did not improve the preference model.

## RESULTS

As expected, our observations suggested that fish did not spend much time in the small compartments. Fish generally entered the smaller compartments and then rapidly returned to the larger compartment. Despite being tested in groups, there was still a high degree of individual variation within each group, suggesting that that the presence of other fish with similar stress profiles did not exacerbate or dampen stress-driven behaviours, and that the results from this study could largely be interpreted on an individual level. However, the random factor (group ID) varied significantly for both models (RLRT analysis $P > 0.05$). This effect was likely due to the idiosyncratic behaviour of individuals in small groups, and thus generalized linear models were run without group as a random factor. Tag detections showed no peak activity time indicating that, contrary to expectations, fish entered the compartments consistently throughout the day and night.

The number of recordings per day increased over time, suggesting that the fish became comfortable with exploring the compartments (Table 1; Fig. 3). Low stress fish entered both compartments significantly more often than high stress fish (Table 1; Fig. 3), which is indicative of higher activity levels over the entire experimental period. Both groups of fish had higher detections for the rewarded compartment overall, suggesting a general preference for the rewarded compartment over the experimental period. There was a significant interaction between day and stress type, indicating that the number of recordings for low stress fish increased faster than high stress fish.

Preference for the reward compartment increased significantly for both treatments over the duration of the experiment, suggesting fish learnt where the reward compartment was (Table 2; Fig. 4). Low stress fish had a significantly higher preference for the reward compartment over the duration of the experiment, though both stress types started and ended at similar levels of preference. Low stress fish achieved their maximum preference at ∼7 days, whereas high responding fish took 12 days to reach a similar level.

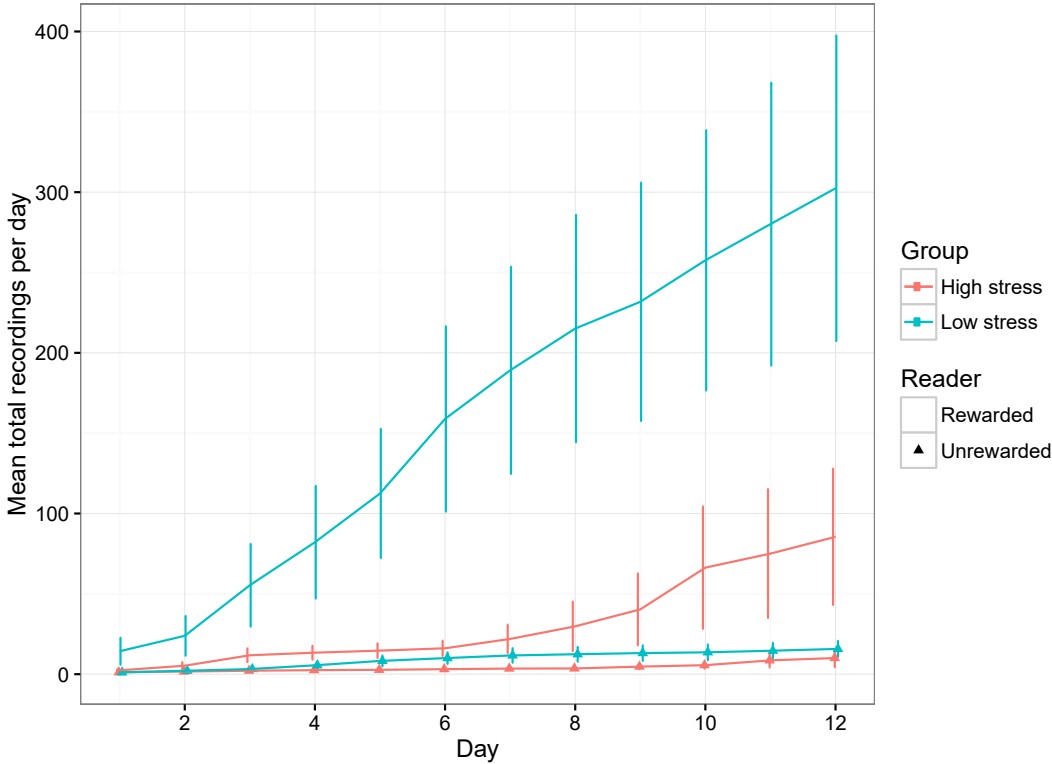

**Figure 3** The mean (±1 SE) number of entries into the rewarded and unrewarded room for low stress and high stress mulloway over time.

**Table 2** Generalized linear model of preference for the rewarded over the unrewarded compartment between low stress and high stress responding fish during the experiment.

| Factors | Estimate | Std. Error | T value | P value |
|---|---|---|---|---|
| Day | 0.12 | 0.03 | 4.25 | <0.001 |
| Stress type | 1.18 | 0.20 | 5.85 | <0.001 |

## DISCUSSION

Our results show that fish with high and low levels of background cortisol had different approaches to the learning task. Low stress fish had much higher levels of activity and entered the rewarded compartment in the simple maze more frequently than high stress fish. Moreover, the preference for the reward room over the unrewarded room showed that low stress fish initially had similar preference as high stress fish but rapidly improved, especially over the first week of the experiment. This suggests that the high activity levels of these fish were tempered by poor accuracy, but they eventually learned the location of the food reward as a result of their pro-active coping style. In contrast, high stress fish had lower levels of activity than low stress fish and rarely entered compartments, but when they did so it was most often the one containing the food reward. Interestingly, their preference was similar to low stress fish to begin with but did not improve to the same extent as low stress fish, which is reminiscent of reactive coping styles (*Koolhaas et al., 1999*). While both

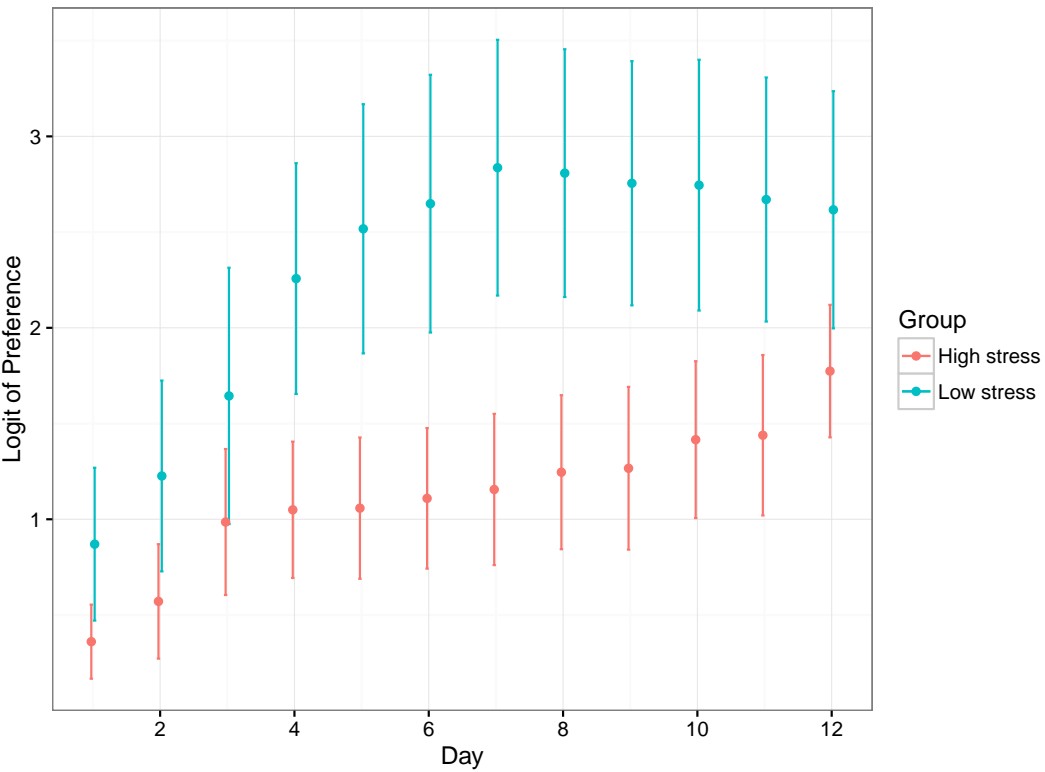

**Figure 4** **Mean (±1 SE) logit-transformed preference for the rewarded compartment in low stress and high stress mulloway over time.** Preference for the rewarded compartment over the unrewarded compartment was used as a proxy for learning in this experimental setup: preference for the rewarded compartment was expected to increase over time as fish learned that food was available in only one of the two compartments. Here, a logit value of 0 indicates no preference for either compartment, and logit values above that indicate greater preference for the rewarded compartment. A logit of 4.5 represents near ~100% preference for the rewarded compartment.

groups attained the same level of performance by the end of the experimental period (day twelve), they arrived at this point at different speeds.

Our results echo work focussing on boldness-accuracy trade-offs (*Mamuneas et al., 2015*). The generally higher levels of activity in our low stress fish enabled them to rapidly explore their environment, placing them in novel contexts and increasing the likelihood of performing innovative behaviours (*Schjolden et al., 2006*). However, this behaviour was offset by the fact that they seemed to pay little attention to appropriate cues and initially suffered from low accuracy, similar to patterns observed in rainbow trout during a reversal task (*Ruiz-Gomez et al., 2011*). High stress fish, by contrast, paid more attention to environmental cues and had a similar initial preference despite lower activity levels.

Assessing stress in this instance required the use of a method approved by an Australian animal ethics committee, as blood samples need to be extracted and a PIT tag implanted into large and powerful fish, a process that, without anaesthesia, would likely be excessively painful for the animals and difficult for handlers. AQUI-S is a widely-used clove oil concentrate that is effective at incapacitating fish in aquaculture environments while being safe for human consumption. It is possible that the use of such an anaesthetic

could interfere with the stress response, however, the use of AQUI-S in *A. regius,* a close relative of *A. japonicus,* revealed that AQUI-S either had no effect on blood plasma cortisol concentrations (*Barata et al., 2016*), or doubled blood plasma cortisol concentrations relative to a control group from 20 to 40 ng/ml (*Cárdenas et al., 2016*). Our previous work identified a positive relationship between sampling order and cortisol blood concentration when subjects were maintained in low concentrations of anaesthetic prior to sampling, but this only explained 6% of the variation (*Raoult et al., 2012b*). Our data suggest that exposure to low concentrations of AQUI-S had little impact on the cortisol levels in this species.

The range of blood cortisol concentrations we detected in *A. japonicus* (10–600 ng/ml) appears to be large. No comparable study exists to examine whether this range is surprising or typical for this species. Studies that examined blood cortisol concentrations in a species from the same genus (*A. regius)* found maximum concentrations of ∼40 ng/ml post mortem (*Millán-Cubillo et al., 2016*) or 269 ng/ml after exposure to clove oil (*Cárdenas et al., 2016*). The *Cárdenas et al. (2016)* study is the most comparable, both in terms of experimental treatment and the range of values observed. Note, however, that in *Cárdenas et al. (2016)* fish were lightly anesthetised using clove oil and placed in fresh seawater for 30 min prior to the blood sample be taken. Their fish were also substantially smaller than ours (136 ± 9 g compared to 495 ± 48.5 g). Larger fishes are known to produce larger stress responses than smaller individuals (*Fatira, Papandroulakis & Pavlidis, 2014*). Future studies should attempt to gauge the scale and speed of blood cortisol responses in *A. japonicus* following a stressor and examine some of the potential causes for the range of background stress levels (e.g., hierarchies; *Colléter & Brown, 2011*).

Populations of fish bred in captive conditions generally have behavior that is skewed towards the bold/low stress side of the behavioral spectrum (*Sundström et al., 2004*; *Kelley, Magurran & García, 2006*). The subjects used herein were bred from wild broodstock, which may explain the wide variation in coping styles observed between individuals. One would expect such variation to be eroded over time through artificial selection given that these physiological traits are known to be heritable (*Benus et al., 1991*; *Koolhaas et al., 2010*). It should thus be feasible to selectively manage coping styles in an aquaculture context if desired, but one must be mindful of altering non-target behavior in the process. Other studies have affected the plasma cortisol responses of a cultured fish (rainbow trout) through selective breeding (*Pottinger & Carrick, 1999*), and it should be possible in other species of fishes (*Pottinger & Pickering, 2011*). One might manage the selection regime depending on whether the fish were destined to be released in the wild or destined for the dinner plate. The presence of a syndrome between activity levels, stress responses, personality and coping styles, however, begs the question as to whether any given trait could be selected individually.

Our study, combined with our previous work linking boldness and stress in this population (*Raoult et al., 2012b*), suggests that differences in coping style and personality can have significant influences on cognitive function and information use in aquaculture species (*Moreira, Pulman & Pottinger, 2004*; *Barreto, Volpato & Pottinger, 2006*; *Kurvers et al., 2010*). Mechanistically, this could be explained by findings that corticosteroids at lower levels have a permissive effect on learning but chronic stress and high levels of

circulating cortisol appear to impair memory (*Roozendaal, 2002*). It may also be that animals with different coping styles employ different learning strategies, for example preferring associative learning over systematic search patterns or vice versa (*Mesquita, Borcato & Huntingford, 2015*). Such differences might explain variation in how fish respond to novelty in an aquaculture setting, for example, in the manner in which they interact with self-feeders (*Alanärä, 1996*) or respond to regular cleaning (*Huntingford & Adams, 2005*). Rapid variation in growth and condition often accumulate in aquaculture populations in the absence of regular sorting and it may well be that much of this variation might be related to the coping strategies employed by particular individuals. Lastly, it might be possible to behaviourally type a species or population before bringing it into an intensive aquaculture setting, provided the user is aware of the costs and benefits that are associated with their choice (*Biro, Beckmann & Stamps, 2010*; *Benus et al., 1991*). Such a screening could predict how the species would respond to life in captivity.

In conclusion, our study shows an association between learning approach, personality and stress responsiveness in a marine fish and has highlighted the effects that coping styles can have on the interaction between marine fish and their captive environment. We suggest that both low and high stress phenotypes may be selected for in different contexts and may explain the wide variation observed in natural populations. Animals with a more cautious phenotype and high stress, reactive coping styles with a concomitant greater degree of phenotypic plasticity may be selected for in highly variable environments (for example release into the wild), whereas low stress, proactive fish may have better outcomes in more stable environments such as those in intensive aquaculture (*Koolhaas et al., 2010*; *Cockrem, 2013*).

## ACKNOWLEDGEMENTS

The facilities were graciously provided by NSW Department of Industry and Investment's Fisheries Research Centre at Cronulla, Sydney. Thanks especially to Dave Barker for all his help.

### Funding

This research was funded by the NSW Department of Primary Industry and the Department of Biological Sciences at Macquarie University, Australia. The funders had no role in study design, data collection and analysis, decision to publish, or preparation of the manuscript.

### Grant Disclosures

The following grant information was disclosed by the authors:
NSW Department of Primary Industry.
Department of Biological Sciences at Macquarie University, Australia.

### Competing Interests

The authors declare there are no competing interests.

## Author Contributions

- Vincent Raoult conceived and designed the experiments, performed the experiments, analyzed the data, wrote the paper, prepared figures and/or tables, reviewed drafts of the paper.
- Larissa Trompf wrote the paper, reviewed drafts of the paper.
- Jane E. Williamson conceived and designed the experiments, wrote the paper, reviewed drafts of the paper.
- Culum Brown conceived and designed the experiments, analyzed the data, wrote the paper, reviewed drafts of the paper.

## Animal Ethics

The following information was supplied relating to ethical approvals (i.e., approving body and any reference numbers):

Macquarie University Animal Ethics and Care group approved this research, approval number 2009/008.

## Data Availability

Raw data has been supplied as Supplemental Information 1.

## Supplemental Information

Supplemental information for this article can be found online at http://dx.doi.org/10.7717/peerj.3445#supplemental-information.

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
