# Peer review of "Stress profile influences learning approach in a marine fish"

_PeerJ, doi:10.7717/peerj.3445_

## Round 0.1 · original submission · Major Revisions

· Academic Editor

Major Revisions

Your manuscript has to be improved according to the suggestions given, The discussion has to be reinforced.

Reviewer 1 ·

Basic reporting

Overall, the paper is clearly written, appropriately organised and well-referenced. The underlying behavioural data, but not the plasma cortisol data, are provided as a supplementary file.

See General Comments for the author for specific queries.

Experimental design

The research does fall within the scope of the journal and aims of the paper are clearly stated. This reviewer has some issues with the experimental design, some aspects of which detract from the reader's confidence in the outcomes.

• A sedative with known effects on the stress axis of fish was employed as part of the stressor.

• Blood samples were collected within 5 minutes of the onset of stress, an interval that for most species is too short for a full stress response to have been resolved.

• The range of blood cortisol levels reported for the fish is extraordinarily large (approx. 20 – 600 ng/ml) given the short period of time (< 5 minutes) between initial stimulus and blood collection.

• The stress response of the target species, the mulloway, has not been fully characterised. Without knowledge of the time-course, levels of cortisol in unstressed fish, and the magnitude of the stress-induced cortisol response, interpreting blood cortisol levels obtained from a procedure such as that employed here is difficult.

These points are expanded upon in the General Comments for the author.

Validity of the findings

The behavioural data set seems robust and is discussed appropriately. Overall, the Discussion is well-written.

Comments for the author

The authors have conducted a study to examine differences in behaviour between two groups of mulloway, defined by having either a low or high cortisol response to a stressor.

The results are interesting, and suggest that differences in cortisol responsiveness to a stressor may reflect differences in behaviour and cognition, consistent with those expected of proactive and reactive personality types. However, some aspects of the study design detract from the reader’s confidence in the outcomes. In short, these can be summarised:

• A sedative with known effects on the stress axis of fish was employed as part of the stressor.

• Blood samples were collected within 5 minutes of the onset of stress, an interval that for most species is too short for a full stress response to have been resolved.

• The range of blood cortisol levels reported for the fish is extraordinarily large (approx. 20 – 600 ng/ml) given the short period of time (< 5 minutes) between initial stimulus and blood collection.

• The stress response of the target species, the mulloway, has not been fully characterised. Without knowledge of the time-course, levels of cortisol in unstressed fish, and the magnitude of the stress-induced cortisol response, interpreting blood cortisol levels obtained from a procedure such as that employed here is difficult.

These points and other more trivial issues/queries are listed below.

1. Line 8. It is stated “Groups of fish were formed based on blood cortisol levels”. It would be helpful for the reader at this point to clarify that these are “stress-induced blood cortisol levels”, not baseline cortisol levels.

2. Line 10. Clarity is important in the abstract. The terms “low stress fish” and “high stress fish” (used throughout but first appearing in the abstract) are potentially confusing to a reader – it is not clear initially from the wording that this refers to fish exhibiting a relatively high or low response to a stressor, rather than fish exposed to a less severe or more severe stressor. Suggest a change in wording, throughout the text, to remove the ambiguity. High responding and low responding has been used by other authors in this context.

3. Line 26. Is personality a “mechanism”? Suggest “factor” or something similar instead.

4. Line 26/27. I am not sure that it is correct to describe the bold-shy continuum as a personality trait. The former applies to a population level phenomenon while the latter is an individual characteristic. Suggest instead “Personality traits such as boldness or shyness….”.

5. Line 34/35. In describing that there are individual differences in efficiency of use of self-feeders it is stated “Historically much of this variation was thought to be explained by hierarchy, but it is likely that variation in personality also plays a role”. Is personality itself not a significant factor in determining position within a hierarchy among fish?

6. Line 35. It is stated that “…research is increasingly directed towards behavioural conditioning of hatchery-reared fish as a means of diminishing mortality rates post release…”. However, the references cited in support of this statement are from 1991, 2000 and 2004. This is not suggestive of an active field of research. Are there more recent examples that support the statement that research in this context is “increasing”?

7. Line 58. …While “a” low stress response may be….

8. Line 60. …such “as” boldness…

9. Line 65. “We have previously shown links between boldness and blood cortisol concentrations in this population…”. Which population? Or do you mean species? Please clarify.

10. Line 68. Suggest “high stress responding” – the term “high stress” alone suggest a state imposed upon the fish.

11. Line 98. Materials and methods. The procedure employed for identifying fish as high responding or low responding are open to question.

In this study blood samples were collected during a minor surgical procedure and the measurement of cortisol in these blood samples was been used to determine whether fish could be classified as high or low responding individuals. The procedure followed is that employed by the authors in a previous study (Raoult et al., 2012. Blood cortisol concentrations predict boldness in juvenile mulloway (Argyosomus japonicus). Journal of Ethology. 30, 225-232) although it is not clear whether these are the same fish as used in the previous study, or a separate population treated identically. (Please clarify)

When comparing the magnitude of stress responses between individuals it is essential that the severity of the stressor, its duration, and the time of blood removal are as similar as possible for all individuals being compared. Here it has been assumed that the handling and disturbance associated with tag insertion constituted a standardised stressor.

Blood samples were collected from fish at the same time as tag insertion occurred and the entire procedure is stated to have required less than 5 minutes from the initial disturbance.

This time period of < 5 mins is surprisingly short to have been used to compare stress responses among fish. When comparing the magnitude of stress responses it is normal practice to allow the response to develop fully, to maximise the levels of cortisol achieved in the blood and for this reason blood collection would normally occur within 30-60 mins after the onset of exposure to the stressor. For most if not all species of fish so far studied the rate of increase of cortisol in the blood is not sufficiently fast that a meaningful response would be evident within 5 minutes of initial disturbance.

However, inspection of Fig. 1 shows the full range of blood cortisol concentrations, measured in mulloway during the tag insertion procedure, which extend from approx. 10 ng/ml to 600 ng/ml. Seemingly, some individuals responded to the procedure, which lasted less than 5 minutes, with virtually no change in cortisol concentration while some increased cortisol levels to several hundred ng/ml. The rapidity of the response implied by the larger concentrations is very different to the rates of change seen in commonly studied taxa.

There appear to be only two previous publications that have examined cortisol levels in mulloway. The previous paper from the present authors, seemingly describing work done with the same population of fish (Raoult et al 2012), and an earlier paper describing effects of capture by angling methods (Broadhurst and Barker, 2000. Archive of Fishery and Marine Research 48, 1-10). The authors of the present paper make no reference to their cortisol data with respect to either of these. It would have been prudent to fully characterise the time-course of the stress response of mulloway before employing blood cortisol levels as a marker for personality type. Given the paucity of information on cortisol levels in this species, and the nature of its stress response, the authors should comment on their cortisol data, preferably with reference to the two preceding papers.

12. Line 86. An additional confounding factor is the inclusion of anaesthesia in the procedure – the effects of anaesthesia/sedation on the stress response in fish are not well understood. It may impair the stress response, or delay it. Certainly, it is a variable that should not have been deliberately incorporated into a stress-testing protocol.

The sedation issue is further compounded because the authors employed a commercially available sedative AQUI-S which contains a clove oil derivative (isoeugenol). Isoeugenol reportedly reduces cortisol concentrations in fish during sedation (Small, B., 2004. Effect of isoeugenol sedation on plasma cortisol, glucose, and lactate dynamics in channel catfish Ictalurus punctatus exposed to three stressors. Aquaculture 238, 469 – 481). It is surprising that a protocol designed to assess differences in individual stress responsiveness within a population of fish first exposed them to a substance known to interfere with the development of that response. Do the authors have any concerns that their results were in any way affected by the use of this sedative? A reader with any knowledge of AQUI-S will query this aspect of the study so it should be addressed.

13. Line 131. The fish were tested as groups of either low-responding or high-responding individuals. Is it possible that the behavioural characteristics of the fish led to interactions within each group that influenced the outcomes? That is, produced an outcome different to that which would have been seen had the fish been tested individually?

14. Results. Figure legends are currently a single sentence. They need to be more informative and not require the reader to refer to the main text to understand the content of the figures.

15. Discussion. This was on the whole well-written and informative.

·

Basic reporting

This article adds to the growing literature of stress coping styles in fish. Although not completely novel, it helps to understand how stress relates to behavioural outputs involving cognition. The writting is overall clear with a professional English level. The references give an adequate background, although I suggest to clarify the differences between cognition and learning and to be consistent thorough the manuscript, since cognition or learning are not mentioned after the introduction, and the results and discussion are explained mainly on the light of fishes preference for either side of the experimental aquarium, not by the cognitive procesess involving such preferences. On line with this, I suggest to reconsider the title too.
The main structure conforms to PeerJ standards. The figures and tables are sufficient and relevant; however, legends should be more self-explanatory, I suggest that you improve the description ej. table one could read: Summary of generalized linear model of recordings of the preference of mulloway for each compartment per day.
I did not find evidence of raw data.

Experimental design

The research is on line woth the aims and scope of PeerJ, and it was carried out following the local ethical standards. The question is well defined and the methodology well define. However, it is somewhat difficult to understand the experimental setting hence, I suggest you improve the experimental tank schematic representation. Also, clarify why you decided to stop recording the movements at 2000 readings, if your main results are based on differences over time. It would be good to have statistical information that such decision does not change the story. Also, cortisol values and ranges for high and low stress fish will give information to understand that both groups were completely different.

Validity of the findings

I wonder why you did not measured cortisol at the end of the experiment to make sure that the differences between groups remained, since cortisol measurements are only a snapshot of the stress reponse, for one specific pont at the time. Also, it will be interesting to know if the individual weights before and after the test changed. I is known that growth rates vary with coping style (Biro et al, J Anim Ecol. 2006 Sep;75(5):1165-71) and that at least in groups of rainbow trout with the same coping styles it is due to a waste in food since low stress responders tend to waste more food, since they usually engage in aggressive behaviour (Overli et al, Aquaculture. Volume 261, Issue 2, 24 November 2006, Pages 776–781), do you think (or even have evidence for) that the same holds for a schooling fish as the molloway?
Clarify, at least in the discussion why not to evaluate this in groups of mixed physiological phenotypes as it will be more representative of what is happening in the wild.
I did not see a discussion/conclusion on the line of congition or learning, consider to review the title.

Comments for the author

The manuscript is well structured and easy to read, and adds to the literature on stress coping styles in fish.

In general, it will probably add to your discussion/conclusion to review more deeply the references related to the HR-LR stress coping styles in rainbow trout starting with Pottinger, T. G. and Carrick, T. R. (1999). Modification of the plasma cortisol response to stress in rainbow trout by selective breeding. General and Comparative Endocrinology 116, 122-132.


Also check the following paper

Benus, R. F., Bohus, B., Koolhaas, J. M., & Vanoortmerssen, G. A. (1991). Heritable variation for aggression as a reflection of individual coping strategies. Experientia 47, 1008-1019.

---

## Round 0.2 · Minor Revisions

· Academic Editor

Minor Revisions

The manuscript has been improved although it still needs some minor corrections. However, it is important that you double check grammar through the whole paper as some major changes seemed to alter the flow of the information.

Reviewer 1 ·

Basic reporting

no comment

Experimental design

see comments below

Validity of the findings

see comments below

Comments for the author

Line numbering listed here is from the “marked” manuscript.

1. Lines 248-249. “low stress” and “high stress” not altered to “responding”.


2. Legend text. Fig. 3. The legend does not explain what the points denoted as “Reader correct” and “Reader incorrect” represent. The text “The output from the PIT tag recorders suggests….” could be deleted as more appropriate for the Discussion.


3. Legend text. Fig. 4. The legend could better explain how the data plotted on the logit-transformed y-axis translate into compartment preference – what the numbers actually represent. It is more clearly explained in the main text.


4. Use of AQUI-S and stress protocol. Lines 293-298. One of my queries about the methodology employed in this study concerned the use of AQUI-S. The authors have provided some clarification in their rebuttal letter regarding the reasons for adopting the protocol described. In the revised ms they state that:

“Assessing stress in cultured animals requires the use of an approved method of anaesthesia in Australian animal ethics, as blood samples need to be extracted from the fish. AQUI-S is a widely-used clove oil concentrate that is effective at incapacitating fish in aquaculture environments, and was thus used in this study.”

Normally, a stressor would be imposed upon the experimental subject and then at an interval or intervals after the stimulus, the animal would be sedated/anaesthetised in order to collect a blood sample for analysis. The text here suggests that Australian animal welfare regulations state that a stressor cannot be imposed upon a fish unless the fish is already sedated. Is this really the case? The text at present is ambiguous and risks misrepresenting the regulations being discussed. Can the authors please clarify whether no stressors may be applied to an un-sedated or un-anaesthetised animal? Or is it only the collection of blood that must be conducted under sedation? The term “Australian animal ethics” is vague – please provide a reference to the actual legislation that applies.


5. Stressor procedure. In my original comments on the manuscript I had a number of questions about the approach adopted to categorize high- and low-responding fish, and the nature of the cortisol response observed. I found the explanations provided in the rebuttal letter a little difficult to follow.

I queried why a very short interval was employed between imposition of the stressor (tag insertion) and collection of blood (< 5 mins). The authors’ response doesn’t really clarify this. They state “We were interested in measuring a short-term stress response that could also be indicative of baseline stress levels, since obtaining baseline values for these large fish was not possible given the facilities that would be required and the number of individuals. We have clarified this in the manuscript lines 113-114.”

I can find no clarification at lines 113-114. They read either “process from capture to recovery took less than 5 min per fish. The netting process caused minimal disturbance to the rest of the fish in the holding tank, although there” or “shortly after a stress event. Blood samples were analysed for cortisol concentrations with a coat-a-count kit from Diagnostic Products Corporation (Los Angeles, CA,” depending on whether mark-up is shown or not.

Can it really be assumed that a short-term stress response would be indicative of baseline (unstressed?) cortisol levels? Why were baseline levels of interest? There is no mention of baseline levels being of importance to the study aims within the ms.


6. Concerning the unusually large range of cortisol values and rapid elevation post-stress. In my original comments I asked whether there might be any explanation for the very large range of cortisol values (approx. 10-600 ng/ml) seen among the fish after tag insertion. Also, whether there are any existing data (a time-course study for example) for this species that might support such a rapid increase in the magnitude of cortisol levels following a short (5 min) stressor?

The authors response on the one hand suggests that differences in time might account for the range of cortisol values (“It is possible that the fish that were tagged/stressed later on in the procedure accumulated more stress”) but then add that time explained only 6% of variation in cortisol levels. If the primary reason for the very wide range in cortisol levels seen among these fish was their “personality” (as the behavioural data do tend to suggest) then one would expect sample time to have a minimal impact. However, it is the very wide range of values seen across such a short period of time (see Fig. 1) that is surprising. The low- and high-responding fish are separated by a margin of several hundred ng/ml. This is why it would be helpful to have these unusual data discussed in the context of earlier data from this species.

The authors state that copyright issues prevent them from discussing earlier data. I don’t understand why there should be copyright issues in discussing data from a previous study, providing proper attribution is provided. In the same way as any previous work is described and discussed, accompanied with the correct citation. Please include in the Discussion a brief comment on the unusual nature of these cortisol results with respect to the apparent range of values and rate of change of cortisol within the 5 minute interval between stressor and sample, and place them in the context of what is known already about maximum stress-induced cortisol levels in this species. Given that the stress-induced cortisol response is used as the basis for segregating the fish it is important that the reader has confidence that the data reported are plausible.

·

Basic reporting

Some parts need to be double-checked for clarity. See general comments

Experimental design

There are some things about the experimental design that still need to be clarified. See general comments

Validity of the findings

Some ponts need to be justified to strenghten the results and discussions. See general comments

Comments for the author

General comments: Line 31. Personality, defined as consistent differences “in behaviour”
Line 33. You decided to change “cognitive style” for “learning approach” check please the whole document and correct accordingly
Line 35. Eliminate “(“ before 2015
Line 45. as a mean
Lines 52-72. Make the connection between personality and stress coping style so they don’t look like separated ideas
Line 125. You use the term behavioural syndrome, you have not defined it (used it) before.
Line 149. I am still concerned with the fact that you did not measure cortisol or weight after the experiment. In fact according with what is stated in lines 140 and 149, almost seven months passed since the measurements were taken. That’s enough time to find changes if any. Please further justify.
Lines 318-326 Precisely, Pottinger and Carrick 1999 intended to select for traits important for aquaculture but they found out that they were also selecting at the same time for other traits bad for aquaculture. So, this part of the discussion is not well placed.
Do you think that a learning process occurred or the differences that you observed were only due to differences in how individuals perceive the environment or even a combination? See again Benus et al 1991, check also Ruiz-Gomez et al (doi: 10.1016/j.physbeh.2010.11.023)

---

## Round 0.3 · accepted · Accept

· Academic Editor

Accept

Thank you for improving your manuscript which now can be published.